# Designing Multi-Modal Embedding Fusion-Based Recommender

**Anna Wróblewska** [1,2,*] **, Jacek Dąbrowski** [1,*] **, Michał Pastuszak** [1] **, Andrzej Michałowski** [1] **, Michał Daniluk** [1,3] **, Barbara Rychalska** [1,2] **, Mikołaj Wieczorek** [1,3] **and Sylwia Sysko-Romańczuk** [4]

1   Synerise S.A., Giełdowa 1, 01-211 Warsaw, Poland; jack.dabrowski@synerise.com (J.D.); michal.pastuszak@synerise.com (M.P.); andrzej.michalowski@synerise.com (A.M.); michal.daniluk@synerise.com (M.D.); barbara.rychalska@synerise.com (B.R.); mikolaj.wieczorek@synerise.com (M.W.)
2   Faculty of Mathematics and Information Science, Warsaw University of Technology, Koszykowa 75, 00-662 Warsaw, Poland
3   Faculty of Electronics and Information Technology, Warsaw University of Technology, Nowowiejska 15/19, 00-665 Warsaw, Poland
4   Faculty of Management, Warsaw University of Technology, Narbutta 85, 02-524 Warsaw, Poland; sylwia.sysko.romanczuk@pw.edu.pl
*   Correspondence: anna.wroblewska1@pw.edu.pl (A.W.); jack.dabrowski@synerise.com (J.D.)

**Abstract:** Recommendation systems have lately been popularised globally. However, often they need to be adapted to particular data and the use case. We have developed a machine learning-based recommendation system, which can be easily applied to almost any items and/or actions domain. Contrary to existing recommendation systems, our system supports multiple types of interaction data with various modalities of metadata through a multi-modal fusion of different data representations. We deployed the system into numerous e-commerce stores, e.g., food and beverages, shoes, fashion items, and telecom operators. We present our system and its main algorithms for data representations and multi-modal fusion. We show benchmark results on open datasets that outperform the state-of-the-art prior work. We also demonstrate use cases for different e-commerce sites.

**Keywords:** recommendations; machine learning; deep learning; multi-modal representation; data representation; embeddings; data fusion

## 1. Introduction

Recommender systems aim to suggest relevant items to users. As items here, we mean: movies to watch, texts to read, products to buy, or anything else, depending on industries. Undoubtedly, the systems are present at almost every large e-commerce store or platform, spanning diverse sectors from garments through jewellery to food.

Multiple frameworks and algorithms exist to build recommender systems. The choice of the optimal approach strongly depends on the types of data available, the distributional properties of the data, modalities considered, and business use cases [1–5]. It is usually impossible to adjust existing algorithms to include a new modality of data or a new type of attributes. Hence, a vast majority of existing recommender systems consider only a single kind of interaction, e.g., clicks or purchases; yet, even in this simple scenario, the generalisation of performance to various datasets seems doubtful [6].

Outside of the currently used data range, businesses desire systems based on predictors derived from variables generated through automatic analysis of customers' voices (audio) and observations of how customers interact with the merchant's websites and mobile and offline ecosystems. Unlike existing solutions, it is expected to use these variables in real-time with data from other channels, thus significantly increasing the systems' effectiveness and expanding its functionality. Improving customer behaviour predictive analytics effectiveness is a crucial challenge for many businesses.

This paper describes our innovative recommender system that utilises a multi-modal fusion of multiple interaction types (e.g., clicks, purchases, adding a product to a cart) and

multiple attribute modalities (audio, video, images, text, other behavioural and sequential data). Our system provides a very efficient framework for combining, deploying, and evaluating recommendation algorithms and scenarios utilising rich, multi-modal, and multi-view data sources. First of all, we clarify the requirements for a next-generation recommender system as follows:

- utilising multiple input interaction types (e.g., clicks, purchases, add-to-cart, geo-locations),
- using multiple input attribute modalities (e.g., text, image, video, other),
- ease of adding new back-end algorithms,
- effective deep learning models for visual search, recommendations with and without session information, which outperform state-of-the-art techniques,
- employing specialised techniques to fuse multiple modalities,
- high efficiency and scalability (services architecture),
- convenient infrastructure for model evaluation and performance measurements.

This work presents our contributions to recommender systems: algorithms for efficient graph embedding, visual data embedding, and multi-modal data fusion. We tested our recommendation deep learning networks fed with data transformed with our embedding approaches and their fusion. We also describe how we utilise these algorithms in the system's overall architecture and use cases.

In the following, Section 2 describes our motivation and related work. Then, we describe our recommendation system architecture and the data workflow in Section 3. Then, main features, such as multi-modal embeddings and their fusion technology, are sketched in Sections 4.1 and 4.3, respectively. Subsequently, we present a few tests with state-of-the-art (SotA) benchmarks (Section 5). Finally, we add a description of our interface, recommendation analytics, and a few use cases from our production deployments (Section 6).

## 2. Motivation and Related Work

The recommendation system aims to help users find the products they need, manage their budget efficiently, and make purchase decisions faster. It is usually achieved by showing related offers and recommending similar products to those they have viewed, suggesting the following products to consider or complement a shopping cart [7,8].

For many years, multiple established recommendation algorithms have operated, ranging from simple heuristic-based methods (such as KNNs) through Collaborative Filtering to deep learning architectures [6,9–11].

Different algorithms are helpful in different input data settings, use cases, and scenarios. Common similar items recommendations based on text and numeric data involve preparing suggestions (i.e., other items or actions to take) considering the context of a single item. Personalised recommendations suggest the products considering the context of users' buying preferences and their behavioural profiles (based on long- or short-term history). The system analyses page visits, transactional data, and product feeds (product metadata) to prepare these suggestions. There are also other types of recommendations, i.e., cross-sell, top products, and last seen offers. A set of recommendation scenarios (which we also use as default settings in our system) are shown in Figure 1.

In practice, the recommendation techniques are often mixed depending on the environment and various factors, e.g., vendor domain, website construction, user history, and current season or time of day. They should be adjusted experimentally and measured constantly. Thus, a highly self-adjustable system to the type and modality of data is crucial for coping with many deployments and using recommendation techniques effectively.

A relatively new trend in building recommendation systems is the latent factor-based approach (LF), i.e., manipulating the latent space of deep learning or other models. The first step is to transform the data, uni- or multi-modal ones, into latent space and then utilise algorithms for recommendation purposes. LF is highly efficient in filtering desired information even from high-dimensional and sparse data [12]. Nevertheless, existing LF model-based approaches mostly ignore the user/service additional data and are susceptible

to noises in data. A remedy is a data-characteristic-aware dense latent factor model [13]. Combining a density peaks-based clustering method into its modelling process, it is less sensitive to noises based on the dense LFs extracted from the initially sparse data. Additionally, the current trend in recommender systems incorporates a latent-factor-analysis-based online sparse-streaming-feature selection algorithm to estimate missing data and, after that, select features. It can significantly improve the quality of streaming feature selection [14].

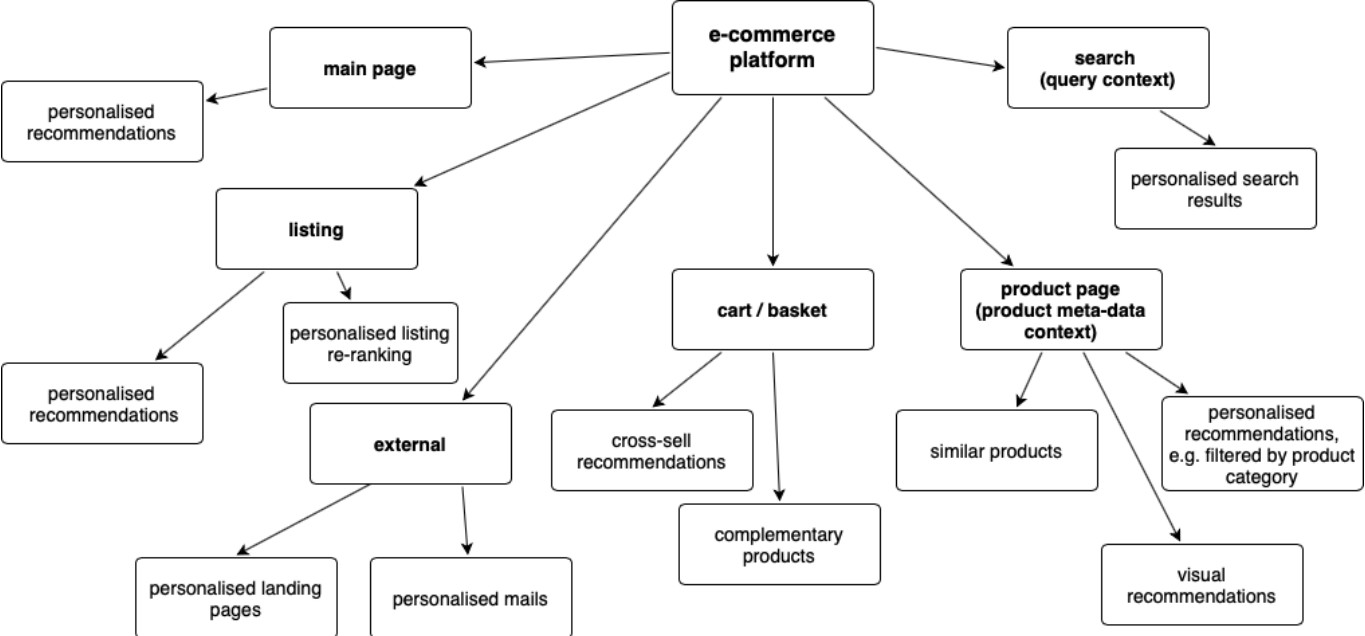

**Figure 1.** Diagram of different recommendation scenarios depending on the site within an e-commerce platform.

Recently, session-based recommender systems (SBRSs) have proven their effectiveness in the recommender systems research area [15]. Content-based and collaborative filtering-based recommender systems usually model long-term and static user selections. SBRSs aim to capture short-term and dynamic user preferences to provide more timely and precise recommendations sensitive to the change in the session contexts. Amongst SBRSs, the item session-based recommenders and the attribute session-based recommenders (ASBRs) utilise the item and attribute session data independently. The feature-weighted session-based recommenders can combine multiple ASBRs with various feature weighting schemes and overcome the cold-start item problem without losing performance [16]. This research and our system comprise algorithms considering users' sessions' long- and short-history and utilising the item and attribute feature space operating in latent space.

### 3. Our System Overview

Our system is prepared to utilise many recommendation scenarios expressed with business rules: different types of recommendations, goals, or filtering expressions. These business rules in recommendation campaigns are consumed by the recommendations facade, together with data from a vendor's platform (taken from its API). Then the business rules written with a dedicated language are parsed. The system runs its logic and obtains or filters item (product) meta-data from the master-item database—dedicated product catalogue. The master-item database items (e.g., product, telecom services meta-data) are kept, along with their attributes and rich data types such as images. The conceptual diagram of the architecture is presented in Figure 2.

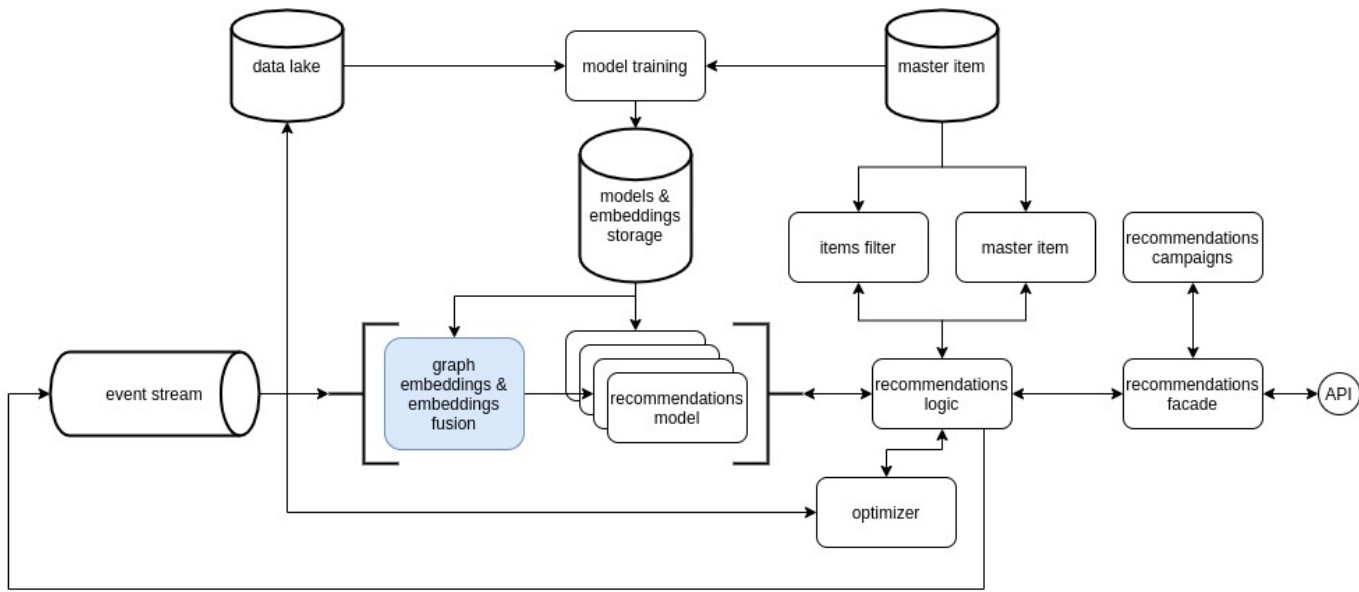

**Figure 2.** A general architecture of our recommendation platform.

Although most of the system works in real-time, the offline part is also present but limited to model training. Algorithms are trained on two primary data sources. The first one is a data lake into which events of different types and origins are being ingested through an events stream; to name a few events types: screen view from a mobile app, product add to cart from a web page, offline transactions, etc. The second source is the above-mentioned master item meta-data database. Of course, the recommendation models also influence the final recommendation logic used to serve recommendations as a request from vendors' platforms triggered by specific user events, e.g., adding a new product to a cart or viewing particular product categories.

## 4. Our Data Embedding and Fusion

### 4.1. Graph Data Embedding

Our algorithms can be fed with various kinds of input data. The system analyses users' long- and short-term interaction history and also item meta-data. For this purpose, we use a multi-step pipeline, starting with unsupervised learning. For images and texts, off-the-shelf unsupervised models may be used. We identify graphs of user-entity interactions (e.g., user-product, user-brand, user-store) and compute multiple graphs or network embeddings for interaction data.

We developed a custom method Cleora for massive-scale network embedding for networks with hundreds of billions of nodes and tens of billions of edges [17]. (The module responsible for the data embedding method is shown in light blue in our system diagram in Figure 2.) The task of network embedding is to map a network or a graph into a low-dimensional embedding space, while preserving higher-order proximities between nodes. In our datasets, nodes represent interacting entities, e.g., users, device IDs, cookies, products, brands, and title words. Edges represent interactions with a single type of interaction per input network, e.g., purchase, view, hover, and search. Thus, the input data is assembled from interacting entities and raw interactions—an edge list for simple- or hyper-graphs.

Similar network embedding approaches include Node2Vec, DeepWalk, and RandNE [18]. These approaches exhibit several undesirable properties, which our method addresses. Thanks to the right design of the algorithm and highly optimised implementation, our method allows for:

- three orders of magnitude improvement in time complexity over Node2Vec and DeepWalk,

- deterministic output—embedding the same network twice results in the same embeddings,
- stable output with regard to small input perturbations—small changes in the dataset result in similar embeddings,
- inductive property and dynamic updating—embeddings for new nodes can be created on the fly,
- applicable to both networks and hyper-networks—support for multi-node edges.

Our custom method works as follows: first, we initialise node vectors (Q matrix) randomly via multiple independent hashing of node labels and map them to the constant interval, resulting in vectors sampled from uniform $(-1, 1)$ distribution. Thus, we achieve deterministic sampling. Empirically, we determine that dimensionality of 1024 or 2048 is enough for most purposes. Then, we calculate a Markov transition matrix (M), representing network connectivity [19]. In the case of hyper-graphs, we perform clique expansion by adding virtual edges. Final node embeddings are achieved by multiplying $M * Q$ iteratively and L2-normalising them in each intermediate step. The number of iterations depends on the distributional properties of the graph, with between three and five iterations being a good default range.

The algorithm is optimised for enormous datasets:

- the Markov transition matrix M is stored in COO (co-occurrence) format in RAM or memory-mapped files on disk;
- all operations are parallelised, concerning the embedding dimensions because dimensions of vectors Q are independent of each other;
- the $M * Q$ multiplication is performed with dimension-level concurrency as well;
- clique expansion for hyper-graphs is performed virtually, only filling the entries in the $M$ matrix;
- star expansion is performed explicitly, with a transient column for the virtual nodes in the input file.

The algorithm's results are entity embeddings contained in the $Q$ matrix. The creation of inductive embeddings (for new nodes) is possible from raw network data using the formula $M' * Q$, where the $M'$ represents the links between existing and new nodes and the $Q$ represents the embeddings of existing nodes.

It is worth noting that the algorithm performs well on interaction networks and short text data, primarily product meta-data. We consider words in a product title as a hyperedge in this setting. It corresponds to star expansion, where product identifiers are virtual nodes linking title words.

### 4.2. Other Embeddings

Our pipeline can efficiently utilise embeddings calculated with the latest techniques of language modelling, e.g., ELMO and BERT embeddings, which are helpful, especially for longer texts. Another data source is visual data (shape, colour, style, etc.), i.e., images. To prepare visual data feed for our algorithm, we use state-of-the-art deep learning neural networks [20,21] customised for our use [22,23]. Indeed, any unsupervised learning method that outputs dense embeddings can be adapted and inputted into our general system pipeline (see Figure 2).

### 4.3. Multi-Modal Fusion

With unsupervised dense representations coming from multiple, possibly different algorithms representing products or other customer entities, we need to aggregate them into fixed-size behavioural profiles for every user.

Our algorithm performs multiple feature space partitionings via vector quantisation [24]. The algorithm involves ideas derived from Locality Sensitive Hashing and Count-Min Sketch algorithm, combined with geometric intuitions. This approach results in sparse representations, which exhibit additive compositionality due to Count-Sketch properties (for a set of items, the sketch of the set is equal to the sum of separate sketches) [25,26].

In our algorithm, all the input modalities and data views (all embedding vectors) are processed independently. Finally, all the results are concatenated into one matrix, called a sketch. The significant advantage of this fusion algorithm is the ability to shorten representations of multiple objects into a much smaller joint representation—the matrix, which allows for easy and fast subsequent retrieval of participating objects in an analogous way to Count-Min Sketch. For example, the user's purchase history can be represented in a single sketch, the website browsing history as another sketch, and the sketches concatenated.

Subsequently, sketches containing user behavioural profiles serve as input to relatively shallow (1–5 layers) feed-forward neural networks. The neural network's output is also designed as a sketch, with the same structure as the input ones.

Training is accomplished with a cross-entropy objective. Output sketches are normalised across the width dimension. During inference, we perform a sketch read-out operation, as in a classic Count-Min Sketch, exchanging the minimum operation to geometric mean, effectively performing averaging of log probabilities [26].

## 5. Experiments on Open Datasets

In our experiments, we tested recommendations in different scenarios: (1) based on visual similarity—retrieving either the same or similar fashion items, (2) user-history-based models for general purpose (not only fashion), (3) feature-based recommendations (based on users' likes and item attributes).

We tested our proprietary deep learning models for recommendations based on visual similarity on big, open datasets commonly used in this field, i.e., DeepFashion and Street2Shop [20,27]. The DeepFashion dataset [28] contains over 800,000 images; we utilised a Consumer-to-shop Clothes Retrieval subset that contains 33,881 unique clothing products and 239,557 images. The Street2Shop dataset [29] comprises over 400,000 shop photos and 20,357 street photos (204,795 distinct clothing items).

On the Street2Shop dataset, we compared our models to two SotA deep learning approaches introduced in [20]. The SotA models comprise two stages: a clothing item detector using the Mask R-CNN detection architecture and dedicated deep learning architecture. We used: (1) a single model—three-stream Siamese network architecture trained with triplet loss and (2) an ensemble model concatenating outputs from the single model and another deep learning, directly optimising the average precision of retrieving images. On the DeepFashion dataset, we compared a deep learning network that was fine-tuned on this training sub-set using the standard triplet loss introduced by [21].

The chosen measures were standard in the employed task of fashion retrieval. They are: mean average precision (mAP) and accuracy at N-th place in the given ranking (Accuracy at N, Acc@N) defined in [20]. In our experiments, we used N = 1, 20, 50, i.e., we measured Acc@1, Acc@20, Acc@50, to compare our results with SotA. Our approach achieved higher results in general and in various garment categories as well (see Table 1 and [22,23]).

**Table 1.** Comparison of performance on our models and published SotA in visual similarity research. The chosen metrics are commonly used in the task [20].

| Dataset—Street2Shop | | | |
|---|---|---|---|
| Model | mAP | Acc@1 | Acc@20 |
| single model [20] | 26.1 | 29.9 | 57.6 |
| ensemble model [20] | 29.7 | 34.4 | 60.4 |
| our custom single model | 37.2 | 42.3 | 61.1 |
| Dataset—DeepFashion | | | |
| Model | Acc@1 | Acc@20 | Acc@50 |
| single model [21] | 27.5 | 65.3 | 76.0 |
| our custom single model | 30.8 | 69.4 | 78.0 |

For a history/session-based recommendations comparison with SotA, we used a framework—testing procedures and datasets published in [30]. Table 2 presents the comparison on two e-commerce datasets, RETAIL and DIGI, containing about 60,000 and 55,000 users' sessions, with a mean number of events per session of 3.54 and 4.78, respectively. The utilised metrics and cutoff point equal to 20 (measures for 20 items—@20—in the recommendation results) also follow the research from [30]. Precision (P) and Recall (R) are counted by comparing the objects of the returned list with the entire remaining session, assuming that not only the immediate next item is relevant for the user. In addition to Precision and Recall, we also report the Mean Average Precision metric (mAP), if the immediate next item is part of the resulting list (Hit Rate, HR), and at which position it is ranked (Mean Reciprocal Rank, MRR).

We compared our model with other SotA methods, such as SKNN, STAN, and VSTNN. The SKNN is a simple session-based nearest-neighbours method that is claimed by research in [30] as a competitor to deep learning in many scenarios. The STAN (called Sequential and Time-Aware Neighbourhood, STAN) is based on the SKNN, taking more information about the users' sessions into account. The VSTAN is an extension and combination of the SKNN and the STAN with a sequence-aware item scoring; it was proposed and proved to be SotA in [30]. Our results are also better or comparable to the results of these methods (see Table 2).

**Table 2.** Comparison of performance on our models and currently published SotA in session-based recommenders research. Metrics are commonly used in the task [30].

| Dataset—RETAIL | | | | | |
|---|---|---|---|---|---|
| Model | mAP@20 | P@20 | R@20 | HR@20 | MRR@20 |
| STAN [31] | 0.0285 | 0.0543 | 0.4748 | 0.5938 | 0.3638 |
| our model | 0.02936 | 0.05405 | 0.48513 | 0.60349 | 0.34454 |
| Dataset—DIGI | | | | | |
| Model | mAP@20 | P@20 | R@20 | HR@20 | MRR@20 |
| SKNN [32] | 0.0255 | 0.0596 | 0.3715 | 0.4748 | 0.1714 |
| VSTAN [32] | 0.0252 | 0.0588 | 0.3723 | 0.4803 | 0.1837 |
| our model | 0.02586 | 0.06037 | 0.37408 | 0.47722 | 0.16645 |

Preliminary results regarding featured recommendations (without user history) show that our proprietary models are comparable to SotA in the field (Precision above 20%—*P*@20; we used the benchmark published in [33], see also [24]). It is much higher than the recent neural approaches, FastAI recommender (achieving only 16.8%), but still lower than Neural Collaborative Filtering (NFC), with 32%. We utilised for this test the MovieLens 20M, as it is the most extended version from the MovieLens datasets [34]; it contains information about almost 140,000 users, giving over 20 million ratings for over 27,000 movies.

Indeed, our algorithms offer significant speed benefits over other neural competitors. For example, our models on the MovieLens dataset take 20 s to train and 14 s to return predictions for 6000 users and 4000 movies (around 23,000,000 user/movie combinations in total). It is faster than recent neural approaches: FastAI recommender or NFC. Accordingly, our method achieves comparable results with [33], using the same hardware (see Table 3).

**Table 3.** Time comparison [s, in seconds]: training and prediction times for data of about 6000 users and 4000 movies (around 23,000,000 user/movie combinations in total).

| Approach | Training Time [s] | Prediction Time [s] |
|---|---|---|
| Ours | 20 | 14 |
| FastAI recommender [35] | 901 | 57 |
| Neural Collaborative Filtering, NCF [36] | 790 | 50 |

## 6. Use Cases

In current production deployments in A/B tests, our platform achieves 20–30% improvements in average order size (AOS) and 10–60% improvements in average order value (AOV) in comparison to the system without our custom deep learning-based recommendations. Of course, the numbers vary significantly depending on the quality of product and user data, recommendation visibility, and vendors' website structure. Figures 3–5 show our different recommendation scenarios in various product categories, i.e., visually similar products and personalised recommendations based on user interactions in various e-commerce platforms. Table 4 provides an example of data about user history and recommendations in the electronics category. Figures 6 and 7 illustrate our recommendation analytics, which provide a customised interface to show aggregated results.

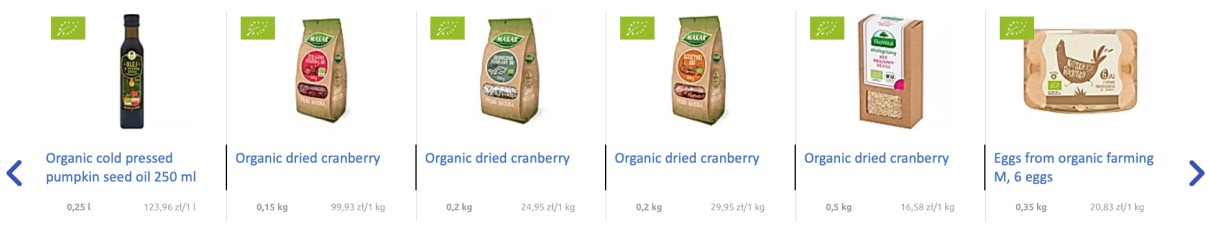

**Figure 3.** Recommendation box at main page of a service—personalised on user interactions in the same session. (The figure is from a Polish vendor, so there is a comma instead of a dot in numbers).

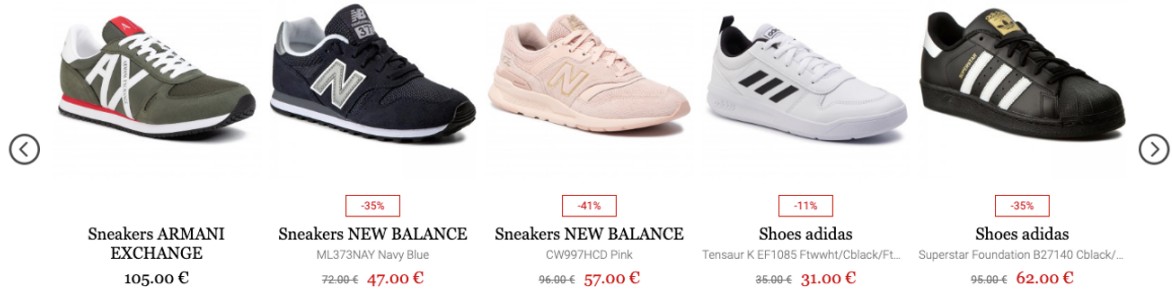

**Figure 4.** Recommendation box—cold start in a session based on previous user behavioural history.

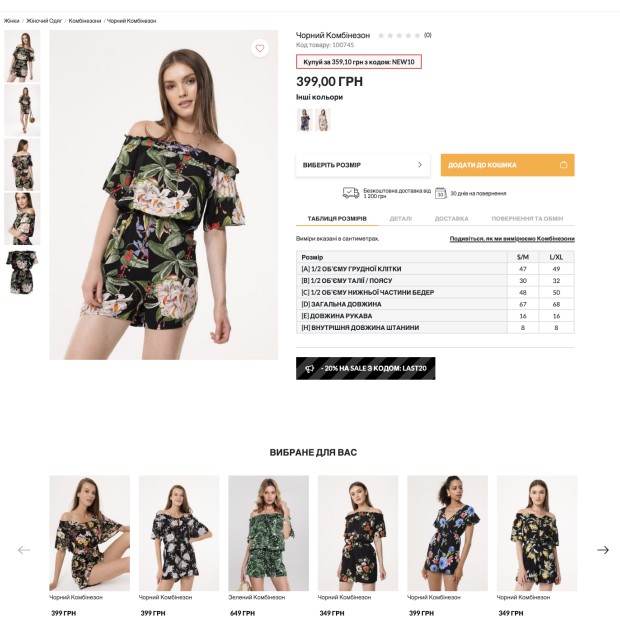

**Figure 5.** Recommendation box—visually similar dresses. (The figure is from a Polish vendor, so there is a comma instead of a dot in numbers).

**Table 4.** Examples of recommendations for our personalised recommenders in electronic products. We can also see complementary products that fit very well to the viewed product.

---

**User history:**
Page Visit: HUAWEI P20 Lite Smartphone Pink
Page Visit: HUAWEI P20 Lite Smartphone Pink
**Our recommendations:**
HUAWEI P20 Lite Smartphone Pink
HUAWEI P20 Lite Smartphone Black
HUAWEI P20 Lite Blue Smartphone
MYSCREEN Lite Edge tempered glass for Huawei P20 Lite Black
HUAWEI Transparent case for Huawei P20 Lite Transparent
MERCURY Jelly Case for Huawei P20 Lite Transparent
HAMA Crystal Clear Cover case for Huawei P20 Lite Transparent
HAMMER case for Huawei P20 Lite Black
HUAWEI Smart Cover for Huawei P20 Lite Black
**User really bought later:** HUAWEI P20 Lite Smartphone Pink

---

**User history:**
Transaction: KARCHER RM 500 cleaner for 500 mL glass
**Our recommendations:**
KARCHER WV 5 Premium window cleaner 1.633–453.0
KARCHER WV Classic 1.633–169.0 window washer
KARCHER WV 2 Premium window cleaner 1.633–430.0
KARCHER RM 500 cleaner for 500 mL glass
KARCHER telescopic lance for WV window washer
KARCHER cleaner RM 503 500 mL
KARCHER microfiber pad for WV window washer (2 pieces)
KARCHER cleaner in 500 mL canister (RM561)
**User really bought later:** KARCHER WV 5 Premium window cleaner 1.633–453.0

---

**User history:**
Page Visit: Smartphone APPLE iPhone 11 Pro Max 256 GB Space Gray
Page Visit: Fridge HAIER A3FE737CMJ
Page Visit: Washing machine AEG LTX7E272P
Page Visit: Smartphone APPLE iPhone 11 Pro Max 256 GB Space Gray
Page Visit: Smartphone APPLE iPhone 11 Pro Max 256 GB Night green
Page Visit: Smartphone APPLE iPhone 11 Pro Max 256 GB Space Gray
**Our recommendations:**
Smartphone APPLE iPhone 11 Pro Max 256 GB Star Gray
Smartphone APPLE iPhone 11 Pro Max 256 GB Night green
Smartphone APPLE iPhone 11 Pro Max 64 GB Star gray
HOFI Glass Pro + tempered glass for Apple iPhone 11 Pro Max
Hybrid glass HOFI Hybrid Glass for Apple iPhone 11 Pro Max Black
APPLE Silicone Case for iPhone 11 Pro Max Black
APPLE Leather Case for iPhone 11 Pro Max Black
SPIGEN Neo Hybrid case for Apple iPhone 11 Pro Max Navy-silver
Watch Dogs 2 Game PS4
**User really bought later:** Smartphone APPLE iPhone 11 Pro Max 256 GB Star Gray

---

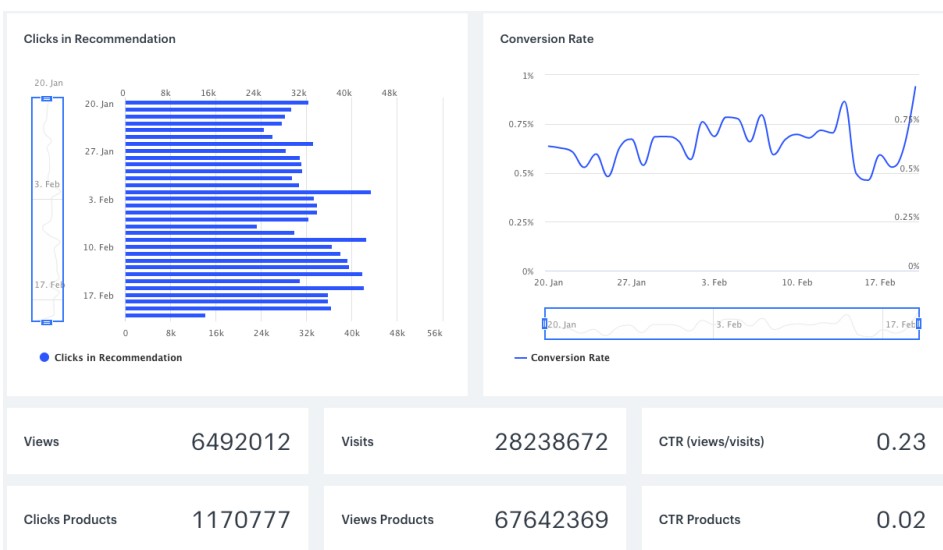

**Figure 6.** Recommendation analytics interface—aggregated results.

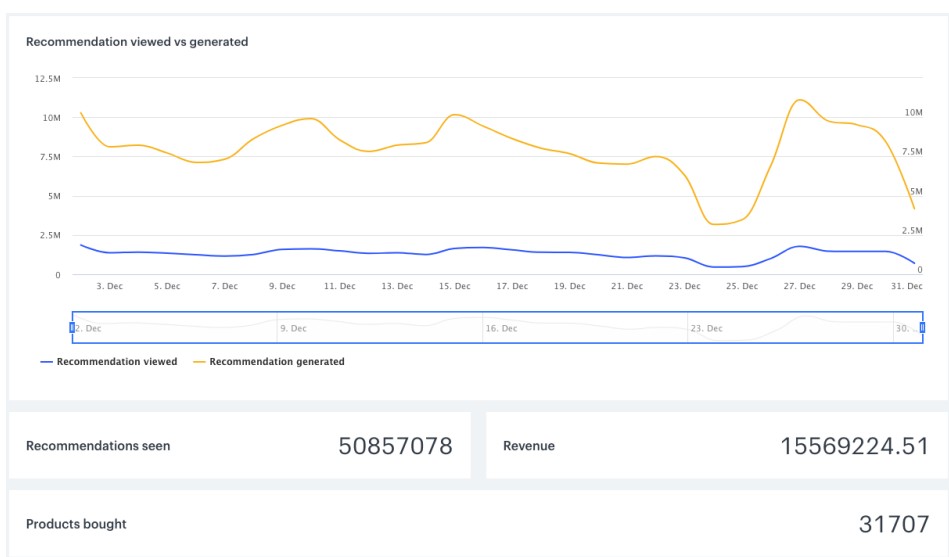

**Figure 7.** Recommendation analytics plot—viewed and clicked recommenations.

## 7. Conclusions

This paper presents our custom recommendation system, our algorithms for preparing data representations and their fusion for multi-modal, multi-view data; and deep-learning-based recommendation models. Our sketch representation for multi-modal data can be applied to any embeddings learned in an unsupervised way, allowing for compact representation with additiveness. Our recommendation algorithms achieved high results across multiple e-commerce stores, and we exceeded state-of-the-art results on open recommendation datasets. Additionally, we illustrate use cases of our system with different scenarios and data feeds. Incidentally, deploying our system in a new e-commerce store takes about one workday, thanks to a modular architecture easily adaptable to clients' APIs and data feeds of different formats.

Future work focuses on product propensity models, demand forecasting, improved search personalisation, and recommendation of non-product entities (e.g., coupons, offers, and brands).

**Author Contributions:** Conceptualisation, A.W., J.D., M.P., A.M. and B.R.; methodology, A.W., J.D. and A.M.; software, J.D., A.M., B.R., M.D. and M.W.; validation, J.D., A.M., M.P., M.D., B.R. and M.W.; formal analysis, J.D., M.P., M.D., A.M., B.R., M.W., A.W., and S.S.-R.; investigation, J.D., A.M., M.D., B.R., M.W. and A.W.; resources, J.D.; data curation, J.D., A.M., M.P., B.R., M.D. and M.W.; writing—original draft preparation, A.W., J.D., A.M., M.P. and B.R.; writing—review and editing, A.W. and S.S.-R.; visualisation, M.P., A.M., M.D. and A.W.; supervision, A.W. and J.D.; project administration, J.D. and A.W.; funding acquisition, J.D., S.S.-R. and A.W. All authors have read and agreed to the published version of the manuscript.

**Funding:** This research was funded by Synerise Ltd. and the National Centre for Research and Development (Narodowe Centrum Badań i Rozwoju) in Poland within the EU co-funded Smart Growth Operational Programme 2014-2020—grant number POIR.01.01.01-00-0695/19.

**Institutional Review Board Statement:** Not applicable.

**Informed Consent Statement:** Not applicable.

**Data Availability Statement:** Open datasets used in our research: Street2Shop—https://papers-withcode.com/dataset/exact-street2shop; DeepFashion—https://liuziwei7.github.io/projects/Deep-Fashion.html; RETAIL and DIGI datasets—https://github.com/rn5l/session-rec; MovieLens—https://grouplens.org/datasets/movielens/; all the links were accessed on 26 January 2022.

**Acknowledgments:** The authors would like to thank the entire Research Artificial Intelligence team and other Machine Learning engineers from Synerise for joint research experiments in production and substantive technological support.

**Conflicts of Interest:** The authors declare no conflict of interest.

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
