# Peer review of "Designing Multi-Modal Embedding Fusion-Based Recommender"

_electronics, doi:10.3390/electronics11091391_

Round 1
Reviewer 1 Report
In e-commerce, it is essential to develop a recommender system suitable for product characteristics such as fashion and food and beverage. Therefore, it is a very interesting and meaningful attempt for the authors to improve the performance of the recommender system using multi-modal embedding.
However, the author needs to improve the following points. If the author accepts them, I believe that this paper will be a good paper that meets the requirements of Electronics.
First, the authors need to rewrite the introduction and conclusions more clearly for the readers of this journal.
Second, the authors need to note recent studies related to session-based recommenders.
i.e. doi.org/10.1145/3465401
doi.org/10.3390/su11123336
Third, it is necessary to reduce the amount of Chapter 3. The authors have to deliver the core content to the readers by deleting unnecessary content.
Fourth, the authors need to assert the importance of this study by emphasizing the theoretical contributions and practical implications of this study.
Author Response
The authors appreciate the comments and suggestions. Thank you very much for your valuable comments and suggestions to improve our paper.
We have thoroughly revised the manuscript according to reviewers’ comments and responded to each comment, point by point. As suggested, the answers to Reviewer comments and suggestions to the author's amendments were introduced to our paper.
Additionally, we have improved and revised the whole text and added many language corrections.
Reviewer’s remark: First, the authors need to rewrite the introduction and conclusions more clearly for the readers of this journal.
Author response: We have rewritten the introduction and conclusions to clarify our contribution and background.
We have added the following paragraph:
Introduction:
“This work presents our contributions to recommender systems: algorithms for efficient graph embedding, visual data embedding and multi-modal data fusion. We tested our recommendation deep learning networks fed with data transformed with our embedding approaches and their fusion. We also describe how we utilise these algorithms in the system's overall architecture and use cases.”
Conclusions:
This paper presents our custom recommendation system, our algorithms for preparing data representations and their fusion for multi-modal, multi-view data, and deep-learning-based recommendation models. Our sketch representation for multi-modal data can be applied to any embeddings learned in an unsupervised way, allowing for compact representation with additiveness. Our recommendation algorithms achieved high results across multiple e-commerce stores, and we exceeded state-of-the-art results on open recommendation datasets.
Reviewer’s remark: Second, the authors need to note recent studies related to session-based recommenders. i.e. doi.org/10.1145/3465401, doi.org/10.3390/su11123336
Authors’ response: We have supplemented the existing literature review with more than five new items and added the following paragraphs.
“A relatively new trend in building recommendation systems is the latent factor-based approach (LF), i.e. manipulating the latent space of deep learning or other models. The first step is to transform the data, uni- or multi-modal ones, into latent space and then utilise algorithms for recommendation purposes.
LF is highly efficient in filtering desired information even from high-dimensional and sparse data~\cite{9411671}. Nevertheless, existing LF model-based approaches mostly ignore the user/service additional data and are susceptible to noises in data. A remedy is a data-characteristic-aware dense latent factor model~\cite{9159907}. Combining a density peaks-based clustering method into its modelling process, it is less sensitive to noises based on the dense LFs extracted from the initially sparse data. Also, the current trend in recommender systems incorporates a latent-factor-analysis-based online sparse-streaming-feature selection algorithm to estimate missing data and, after that, select features. It can significantly improve the quality of streaming feature selection~\cite{9506984}.
Recently session-based recommender systems (SBRSs) have proved their effectiveness in the recommender systems research area~\cite{10.1145/3465401}. Content-based and collaborative filtering-based recommender systems usually model long-term and static user selections. SBRSs aim to capture short-term and dynamic user preferences to provide more timely and precise recommendations sensitive to the change in the session contexts. Amongst SBRSs, the item session-based recommenders and the attribute session-based recommenders (ASBRs) utilise the item and attribute session data independently. The feature-weighted session-based recommenders can combine multiple ASBRs with various feature weighting schemes and overcome the cold-start item problem without losing performance~\cite{sbr-2019}.
This research and our system comprise algorithms considering users' long- and short-history sessions and utilising the item and attribute feature space operating in latent space.”
BibTex
@article{10.1145/3465401, author = {Wang, Shoujin and Cao, Longbing and Wang, Yan and Sheng, Quan Z. and Orgun, Mehmet A. and Lian, Defu}, title = {A Survey on Session-Based Recommender Systems}, year = {2021}, issue_date = {September 2022}, publisher = {Association for Computing Machinery}, address = {New York, NY, USA}, volume = {54}, number = {7}, issn = {0360-0300}, url = {https://doi.org/10.1145/3465401}, doi = {10.1145/3465401},
@article{sbr-2019, title={Session-Based Recommender System for Sustainable Digital Marketing}, volume={11}, ISSN={2071-1050}, url={http://dx.doi.org/10.3390/su11123336}, DOI={10.3390/su11123336}, number={12}, journal={Sustainability}, publisher={MDPI AG}, author={Hwangbo, Hyunwoo and Kim, Yangsok}, year={2019}, month={Jun}, pages={3336} }
@ARTICLE{9411671, author={Wu, Di and Shang, Mingsheng and Luo, Xin and Wang, Zidong}, journal={IEEE Transactions on Neural Networks and Learning Systems}, title={An L₁-and-L₂-Norm-Oriented Latent Factor Model for Recommender Systems}, year={2021}, volume={}, number={}, pages={1-14}, doi={10.1109/TNNLS.2021.3071392}}
@ARTICLE{9159907, author={Wu, Di and Luo, Xin and Shang, Mingsheng and He, Yi and Wang, Guoyin and Wu, Xindong}, journal={IEEE Transactions on Knowledge and Data Engineering}, title={A Data-Characteristic-Aware Latent Factor Model for Web Services QoS Prediction}, year={2020}, volume={}, number={}, pages={1-1}, doi={10.1109/TKDE.2020.3014302}}
@ARTICLE{9506984, author={Wu, Di and He, Yi and Luo, Xin and Zhou, MengChu}, journal={IEEE Transactions on Systems, Man, and Cybernetics: Systems}, title={A Latent Factor Analysis-Based Approach to Online Sparse Streaming Feature Selection}, year={2021}, volume={}, number={}, pages={1-15}, doi={10.1109/TSMC.2021.3096065}}
Reviewer’s remark: Third, it is necessary to reduce the amount of Chapter 3. The authors have to deliver the core content to the readers by deleting unnecessary content.
Author response:
After your remark and further consideration, we have decided to remove the implementation details of our system and the part on language to define recommendation business rules. This part, which is an innovation in our system, is not in the general scope of this paper and research.
To better structure our paper, we divided the 3rd section into “Our system overview” and our methods - “Our Data Embedding and Fusion”. Additionally, we have improved the description of our methods and added a few background literature references.
Thus, we have removed the following texts:
“The system is based on Reactive Microservices Architecture~\cite{RMA_2016,rmanifesto}, implementing its core principles which are: elasticity, scalability, fault tolerance, high availability, message-driven, and real-time processing. Not only are scores and recommendations calculated during the request time, but user representations are also being updated and exposed to models after each event flowing through the event stream. Especially real-time processing is crucial to provide tailored, high-quality recommendations considering the latest changes in in-session user behaviour and changes in system performance. Not only are scores and recommendations calculated during the request time, but user representations are also being updated and exposed to models after each event flowing through the event stream.”
“IQL custom query language provides a flexible framework for building new recommendation scenarios based on item meta-data and recommendation request context. In Fig.~\ref{fig:iql} there are a few examples of making recommendation filtering rules. IQL expressions are handled by an items filter, which performs filtering of candidate items based on given constraints. To achieve high throughput and low latency, the items filter uses its compressed binary representation of items, serving thousands of requests per second and filtering sets of million+ items. In the case of IQL expressions with low selectivity, transferring the data structure containing candidate item IDs over the network infrastructure could be expensive; therefore, a binary protocol between filter and logic has been implemented. Optimiser implements a form of a Thompson Sampling algorithm solving multi-armed bandit problems allowing not only to easily A/B test new ideas and algorithms but also to optimise results of running recommendation campaigns. The Optimizer selects the model which will handle the request. Finally, one of the models receives a request to score available candidates based on the model itself and update entity embeddings.”
Reviewer’s remark: Fourth, the authors need to assert the importance of this study by emphasising its theoretical contributions and practical implications of this study.
Author response: We have emphasised our study's contribution and implications in the introduction, conclusions, and experiments.
Please, refer to the first remark. We have also added a more detailed description of the used dataset, the comparison methods, and measures to the Experiments section.
“In our experiments, we tested recommendations in different scenarios: (1) based on visual similarity -- retrieving either the same or similar fashion items, (2) user-history based models for general purpose (not only fashion), (3) feature-based recommendations (based on users' likes and item attributes).”
“The DeepFashion dataset~\cite{7780493} contains over 800,000 images, we utilized a Consumer-to-shop Clothes Retrieval subset that contains 33,881 unique clothing products and 239,557 images. The Street2Shop dataset~\cite{7410739} comprises over 400,000 shop photos and 20,357 street photos (204,795 distinct clothing items).
On the Street2Shop dataset, we compared our models to two SotA deep learning approaches introduced in~\cite{kucer_detect-then-retrieve_2019}. The SotA models comprise two stages: a clothing item detector using the Mask R-CNN detection architecture and dedicated deep learning architecture. We used: a single model -- three-stream Siamese network architecture trained with triplet loss. An ensemble model concatenates outputs from the single model, and another deep learning directly optimises the average precision of retrieving images. On the DeepFashion dataset, we compared a deep learning network that was fine-tuned on this training sub-set using the standard triplet loss introduced by~\cite{dodds_learning_2018}.
The chosen measures were common in the employed task of fashion retrieval. They are: mean average precision (mAP), accuracy at N-th place in the given ranking (Accuracy at N, Acc@N) defined in~\cite{kucer_detect-then-retrieve_2019}. In our experiments, we used N=1, 20, 50, i.e. we measured Acc@1, Acc@20, Acc@50 to compare our results with SotA.“
Reviewer 2 Report
This paper proposes a machine learning-based recommendation platform, which supports multiple types of interaction data with various modalities of metadata. It is achieved through a multi-modal fusion of various data representations. Experiments on various e-commerce stores datasets show that the proposed platform has advantages over the state-of-the-art work. This paper has some contributions. However, I still have some concerns as follows:
- The background description of the recommender system is insufficient. The recommender system is a well-established technology. However, the related models and methods concerning the recommender system are not enough in this paper.
- The contribution is not clear in this paper. It should illustrate which methods or technologies are proposed in this paper and what experimental results have been achieved.
- The font size in figures and tables is too small, we could hardly read the text on them clearly. Please authors redesign the layout of these figures and tables.
- Table 1 summarizes the experimental results. However, these metrics used in table 1 are not given their description and function. The authors should give more details on this part.
- The time cost is a critical metric. Authors should compare the total time cost of comparison models and summary their total time cost in a table.
- The comparison models used in this paper should be given more details, such as their source and description.
- The datasets are not described in detail in this article. How about their sizes and form? Please rich it.
- Latent factor analysis is one of the most popular techniques to develop a recommender. It is better to discuss recent studies regarding latent factor analysis. For instance, the following papers are relevant and may be useful for this work:
- An L1-and-L2-norm-oriented Latent Factor Model for Recommender Systems, IEEE Transactions on Neural Networks and Learning Systems, DOI: 10.1109/TNNLS.2021.3071392.
- A data-characteristic-aware latent factor model for web services QoS prediction,IEEE Transactions on Knowledge and Data Engineering,2020, doi: 10.1109/TKDE.2020.3014302.
- A Latent Factor Analysis-based Approach to Online Sparse Streaming Feature Selection, IEEE Transactions on Systems Man and Cybernetics-Systems, 2021, DOI: 10.1109/TSMC.2021.3096065
Author Response
The authors appreciate the comments and suggestions. Thank you very much for your valuable comments and suggestions to improve our paper.
We have thoroughly revised the manuscript according to reviewers’ comments and responded to each comment, point by point. As suggested, the answers to Reviewer comments and suggestions to the author's amendments were introduced to our paper.
Additionally, we have improved and revised the whole text and added many language corrections.
Reviewer’s remark: The background description of the recommender system is insufficient. The recommender system is a well-established technology. However, the related models and methods concerning the recommender system are not enough in this paper.
Authors’ response: We have supplemented the existing literature review with more than five new items and added the following paragraphs.
“A relatively new trend in building recommendation systems is the latent factor-based approach (LF), i.e. manipulating the latent space of deep learning or other models. The first step is to transform the data, uni- or multi-modal ones, into latent space and then utilise algorithms for recommendation purposes.
LF is highly efficient in filtering desired information even from high-dimensional and sparse data~\cite{9411671}. Nevertheless, existing LF model-based approaches mostly ignore the user/service additional data and are susceptible to noises in data. A remedy is a data-characteristic-aware dense latent factor model~\cite{9159907}. Combining a density peaks-based clustering method into its modelling process, it is less sensitive to noises based on the dense LFs extracted from the initially sparse data. Also, the current trend in recommender systems incorporates a latent-factor-analysis-based online sparse-streaming-feature selection algorithm to estimate missing data and, after that, select features. It can significantly improve the quality of streaming feature selection~\cite{9506984}.
Recently session-based recommender systems (SBRSs) have proved their effectiveness in the recommender systems research area~\cite{10.1145/3465401}. Content-based and collaborative filtering-based recommender systems usually model long-term and static user selections. SBRSs aim to capture short-term and dynamic user preferences to provide more timely and precise recommendations sensitive to the change in the session contexts. Amongst SBRSs, the item session-based recommenders and the attribute session-based recommenders (ASBRs) utilise the item and attribute session data independently. The feature-weighted session-based recommenders can combine multiple ASBRs with various feature weighting schemes and overcome the cold-start item problem without losing performance~\cite{sbr-2019}.
This research and our system comprise algorithms considering users' long- and short-history sessions and utilising the item and attribute feature space operating in latent space.”
BibTex
@article{10.1145/3465401, author = {Wang, Shoujin and Cao, Longbing and Wang, Yan and Sheng, Quan Z. and Orgun, Mehmet A. and Lian, Defu}, title = {A Survey on Session-Based Recommender Systems}, year = {2021}, issue_date = {September 2022}, publisher = {Association for Computing Machinery}, address = {New York, NY, USA}, volume = {54}, number = {7}, issn = {0360-0300}, url = {https://doi.org/10.1145/3465401}, doi = {10.1145/3465401},
@article{sbr-2019, title={Session-Based Recommender System for Sustainable Digital Marketing}, volume={11}, ISSN={2071-1050}, url={http://dx.doi.org/10.3390/su11123336}, DOI={10.3390/su11123336}, number={12}, journal={Sustainability}, publisher={MDPI AG}, author={Hwangbo, Hyunwoo and Kim, Yangsok}, year={2019}, month={Jun}, pages={3336} }
@ARTICLE{9411671, author={Wu, Di and Shang, Mingsheng and Luo, Xin and Wang, Zidong}, journal={IEEE Transactions on Neural Networks and Learning Systems}, title={An L₁-and-L₂-Norm-Oriented Latent Factor Model for Recommender Systems}, year={2021}, volume={}, number={}, pages={1-14}, doi={10.1109/TNNLS.2021.3071392}}
@ARTICLE{9159907, author={Wu, Di and Luo, Xin and Shang, Mingsheng and He, Yi and Wang, Guoyin and Wu, Xindong}, journal={IEEE Transactions on Knowledge and Data Engineering}, title={A Data-Characteristic-Aware Latent Factor Model for Web Services QoS Prediction}, year={2020}, volume={}, number={}, pages={1-1}, doi={10.1109/TKDE.2020.3014302}}
@ARTICLE{9506984, author={Wu, Di and He, Yi and Luo, Xin and Zhou, MengChu}, journal={IEEE Transactions on Systems, Man, and Cybernetics: Systems}, title={A Latent Factor Analysis-Based Approach to Online Sparse Streaming Feature Selection}, year={2021}, volume={}, number={}, pages={1-15}, doi={10.1109/TSMC.2021.3096065}}
Reviewer’s remark: The contribution is not clear in this paper. It should illustrate which methods or technologies are proposed in this paper and what experimental results have been achieved.
Author response: We have emphasised the contribution and implications of our work in the introduction, conclusions and the Experiments section.
We have added the following paragraph:
Introduction:
“This work presents our contributions to recommender systems: algorithms for efficient graph embedding, visual data embedding and multi-modal data fusion. We tested our recommendation deep learning networks fed with data transformed with our embedding approaches and their fusion. We also describe how we utilise these algorithms in the system's overall architecture and use cases.”
Conclusions:
This paper presents our custom recommendation system, our algorithms for preparing data representations and their fusion for multi-modal, multi-view data, and deep-learning-based recommendation models. Our sketch representation for multi-modal data can be applied to any embeddings learned in an unsupervised way, allowing for compact representation with additiveness. Our recommendation algorithms achieved high results across multiple e-commerce stores, and we exceeded state-of-the-art results on open recommendation datasets.
Reviewer’s remark: The font size in figures and tables is too small, we could hardly read the text on them clearly. Please authors redesign the layout of these figures and tables.
Author response: We have enlarged the figures. We have also proposed a horizontal view for the first two figures. However, this proposition requires the Editors’ approval.
Reviewer’s remark: Table 1 summarises the experimental results. However, these metrics used in table 1 are not given their description and function. The authors should give more details on this part.
Author response: We have described the metrics in all the experiments on the open dataset.
“The chosen measures were common in the employed task of fashion retrieval. They are: mean average precision (mAP), accuracy at N-th place in the given ranking (Accuracy at N, Acc@N) defined in~\cite{kucer_detect-then-retrieve_2019}. In our experiments, we used N=1, 20, 50, i.e. we measured Acc@1, Acc@20, Acc@50, to compare our results with SotA.“
“Table~\ref{tab:sota_comparison_sbr} presents the comparison on two e-commerce datasets: RETAIL and DIGI, containing about 60,000 and 55,000 users' sessions with a mean number of events per session 3.54 and 4.78, respectively. The utilised metrics and cutoff point equal to 20 (measures for 20 items -- @20 -- in the recommendation results) also follow the research from~\cite{ludewig2019empirical}. Precision (P) and Recall (R) are counted by comparing the objects of the returned list with the entire remaining session, assuming that not only the immediate next item is relevant for the user. In addition to Precision and Recall, we also report the Mean Average Precision metric (mAP) and for measuring if the immediate next item is part of the resulting list (Hit Rate, HR) and at which position it is ranked (Mean Reciprocal Rank, MRR). ”
Reviewer’s remark: The time cost is a critical metric. Authors should compare the total time cost of comparison models and summary their total time cost in a table.
Author response: We have summarised the costs and given them in Table 3.
Indeed, our algorithms offer significant speed benefits over other neural competitors. For example, our models on the MovieLens dataset take 20 seconds to train and 14 seconds to return predictions for 6000 users and 4000 movies (around 23.000.000 user/movie combinations in total), which is much higher compared to recent neural approaches: FastAI recommender or Neural Collaborative Filtering (NFC), while achieving comparable results with \cite{microsoft_2019}, using the same hardware (see Table~\ref{tab:time}).
\begin{table}[!ht]
\begin{center}
\caption{Time comparison [sec, in seconds]: training and prediction times for data of about 6,000 users and 4,000 movies (around 23,000,000 user/movie combinations in total)}
\label{tab:time}
\begin{tabular}{l|cc}
\hline
Approach & Training time [sec] & Prediction time [sec] \\
Ours & 20 & 14 \\
FastAI recommender \cite{howard2018fastai} & 901 & 57 \\
Neural Collaborative Filtering, NCF \cite{he_2017} & 790 & 50 \\
\hline
\end{tabular}
\end{center}
\end{table}
Reviewer’s remark: The comparison models used in this paper should be given more details, such as their source and description.
Author response: We have described the comparison models and structuralised the experiments more clearly by completing the description of datasets, metrics and source references.
"On the Street2Shop dataset, we compared our models to two SotA deep learning approaches introduced in~\cite{kucer_detect-then-retrieve_2019}. The SotA models comprise two stages: a clothing item detector using the Mask R-CNN detection architecture and dedicated deep learning architecture. We used: a single model -- three-stream Siamese network architecture trained with triplet loss, an ensemble model concatenating outputs from the single model and another deep learning directly optimising the average precision of retrieving images. On the DeepFashion dataset, we compared a deep learning network that was fine-tuned on this training sub-set using the standard triplet loss introduced by~\cite{dodds_learning_2018}.
We compared our model with other SotA methods, such as SKNN, STAN, and VSTNN. SKNN is a simple session-based nearest-neighbours method that is claimed by research in~\cite{ludewig2019empirical} as a competitive to deep learning in many scenarios. STAN (called Sequential and Time-Aware Neighbourhood, STAN) is based on SKNN taking more information about the users' sessions into account. VSTAN is an extension and combination of SKNN and STAN with a sequence-aware item scoring; it was proposed and proved to be SotA in~\cite{ludewig2019empirical}. Our results are also better or comparable to the results of these methods (see Table~\ref{tab:sota_comparison_sbr}).
Reviewer’s remark: The datasets are not described in detail in this article. How about their sizes and form? Please rich it.
Author response: We have completed the description of datasets and their source references.
“The DeepFashion dataset~\cite{7780493} contains over 800,000 images, we utilized a Consumer-to-shop Clothes Retrieval subset that contains 33,881 unique clothing products and 239,557 images. The Street2Shop dataset~\cite{7410739} comprises over 400,000 shop photos and 20,357 street photos (204,795 distinct clothing items).
Table~\ref{tab:sota_comparison_sbr} presents the comparison on two e-commerce datasets: RETAIL and DIGI, containing about 60,000 and 55,000 users' sessions with a mean number of events per session 3.54 and 4.78, respectively.
“We utilised for this test the MovieLens 20M is the biggest version from MovieLens datasets\cite{MovieLens-data}; it contains information about almost 140,000 users giving above 20 million ratings for above 27,000 movies.”
Reviewer’s remark: Latent factor analysis is one of the most popular techniques to develop a recommender. It is better to discuss recent studies regarding latent factor analysis. For instance, the following papers are relevant and may be useful for this work:
- An L1-and-L2-norm-oriented Latent Factor Model for Recommender Systems, IEEE Transactions on Neural Networks and Learning Systems, DOI: 10.1109/TNNLS.2021.3071392.
- A data-characteristic-aware latent factor model for web services QoS prediction,IEEE Transactions on Knowledge and Data Engineering,2020, doi: 10.1109/TKDE.2020.3014302.
- A Latent Factor Analysis-based Approach to Online Sparse Streaming Feature Selection, IEEE Transactions on Systems Man and Cybernetics-Systems, 2021, DOI: 10.1109/TSMC.2021.3096065
Authors” response: Please, refer to the first remark.
Round 2
Reviewer 1 Report
The authors faithfully reflected the reviewer's requirements in the revised paper. I believe this paper is eligible for the publication in Electronics.
Author Response
Thank you for all your valuable remarks and suggestions. They have allowed us to boost our paper to a considerable extent. Once again, we have thoroughly read the paper and improved our English language.
Reviewer 2 Report
Most of my concerns have been addressed. This paper could be accepted after a minor revision of language
Author Response

(The authors gave the same response as above.)
